# STOCHASTIC NEURAL PHYSICS PREDICTOR

## ABSTRACT

Recently, neural-network based forward dynamics models have been proposed that attempt to learn the dynamics of physical systems in a deterministic way. While near-term motion can be predicted accurately, long-term predictions suffer from accumulating input and prediction errors which can lead to plausible but different trajectories that diverge from the ground truth. A system that predicts distributions of the future physical states for long time horizons based on its uncertainty is thus a promising solution. In this work, we introduce a novel robust Monte Carlo sampling based graph-convolutional dropout method that allows us to sample multiple plausible trajectories for an initial state given a neural-network based forward dynamics predictor. By introducing a new shape preservation loss and training our dynamics model recurrently, we stabilize long-term predictions. We show that our model's long-term forward dynamics prediction errors on complicated physical interactions of rigid and deformable objects of various shapes are significantly lower than existing strong baselines. Lastly, we demonstrate how generating multiple trajectories with our Monte Carlo dropout method can be used to train model-free reinforcement learning agents faster and to better solutions on simple manipulation tasks.

## 1 INTRODUCTION

Learning to predict the physical motion of objects from data is an open area of research. Yet, recent (hierarchical) relation network based forward dynamics predictors (Battaglia et al., 2016; Chang et al., 2016; Mrowca et al., 2018; Li et al., 2019) seem to be a promising alternative to conventional physics engines that are key components of robot control, computer vision and reinforcement learning (RL) systems. Physics simulators, both traditional numerical solvers and learned prediction models, still suffer from insufficient accuracy in challenging scenarios. Small errors in the input and model can lead to dramatically different object trajectories. Take the orange ball that is falling on the blue wedge in Figure 1. Depending on where the orange ball starts or what bias the model has, the ball could either end up on the left or right side. Both are valid outcomes. However, deterministic physics engines will either predict one trajectory or the other.

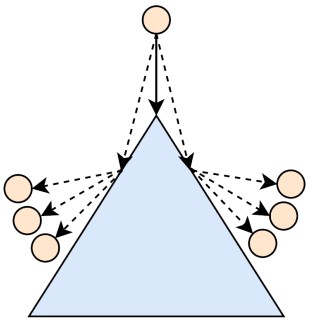

Figure 1: **Uncertainty in physics.** Small errors in the input and prediction can lead to significantly different object trajectories. The orange ball could either end up on the left or right side of the wedge.

While it is important to reduce errors in each prediction, it is also important to acknowledge that uncertain situations might not have one but multiple possible outcomes. In machine learning, uncertainty-aware neural networks avoid deterministic point estimates by predicting distributions or by randomly sampling in the prediction interval. In the context of dynamics predictions, we propose to use Monte Carlo sampling based dropout on the model weights of a learned forward dynamics predictor to model uncertainty and sample multiple plausible trajectories for an initial state. To stabilize each trajectory and reduce error accumulation over long-time horizons, we use a state-invariant recurrent training mechanism. By feeding back predictions as input over multiple time steps, the model becomes more robust to its own prediction errors without the need for a hidden state. Finally, we introduce a new shape loss on the model predictions that constrains the pairwise

distances between objects and object parts and greatly improves shape preservation and the stability of trajectories over long-time horizons. Our final fully differentiable forward dynamics model is able to sample multiple, more accurate and more stable trajectories over long-time horizons compared to existing baselines.

An accurate forward dynamics predictor that is able to predict a distribution of future states can be of great importance for robotic control. In model-free reinforcement learning, accomplishing tasks through random exploration is sample inefficient and hardly generalizable. Model-based methods promise greater generalization abilities, but suffer from deterministic world models that are hard to learn and fail in stochastic environments. With our stochastic forward dynamics predictor, we can move part of the sampling process into the environment, physically grounding the random exploration of model-free agents. As the agent is able to observe multiple trajectories at a given state without actually executing multiple actions, the sample efficiency is greatly improved while the stochasticity of each state and action is implicitly learned. We show on several control experiments that a model-free agent trained in our stochastic forward dynamics environment is not only able to better explore and learn faster but often also comes to better solutions than agents trained in deterministic environments.

In summary, (1) we propose a stochastic differentiable forward dynamics model that is able to generate multiple plausible trajectories via Monte Carlo (MC) based graph-convolutional dropout. (2) We greatly improve the accuracy and stability of long-term predictions by proposing a new fully-connected shape loss term and training the model recurrently end-to-end in a state-invariant way. (3) We demonstrate how our stochastic dynamics model can be used to improve the efficiency and performance of model-free reinforcement learning agents on several physical manipulation tasks.

## 2 RELATED WORK

Physical dynamics prediction has long been an open research questions (Fragkiadaki et al., 2015; Agrawal et al., 2016; Li et al., 2016; Finn et al., 2016; Lerer et al., 2016; Mottaghi et al., 2016a;b; Haber et al., 2018; Tran et al., 2015; 2016; Qi et al., 2017a;b; Byravan & Fox, 2017). Recent advancements in deep learning allowed for emergence of successful systems that aim at solving this problem by learning from data. Battaglia et al. (2016) and Chang et al. (2016) proposed a graph-based approach with object-centric and relation-centric representations, and a neural network architecture that predicts object dynamics and interaction between objects in complex 2D scenes. Similarly, Mrowca et al. (2018) and Li et al. (2019) implement a relational network with particle representation for objects, but extend this approach to 3D scenes introducing hierarchical graph representations for computational tractability. These works rely however on a single step prediction during training. We propose a recurrent training scheme on multiple step predictions and show lower long-term error in our experiments.

Simulating future plausible object states under physical and user constraints is a commonly addressed challenge in computer graphics. Chenney & Forsyth (2000) analysed uncertainties in their simulation model for multi-body scenarios and used a Markov chain Monte Carlo algorithm to predict multiple trajectories. Twigg & James (2007) based their work on psychological findings about human errors in predicting object dynamics and simulated multiple future environment states by applying external random impulses to colliding bodies. Trajectories generated by both of the mentioned methods are visually plausible to human, but can often diverge from the real physical behavior and require extensive expertise to choose simulation parameters that ensure convergence. Huberman & Struss (1992) support the psychological theory behind the latter work, further suggesting that human predictions of non-linear dynamic effects such as collisions are far from perfect, and thus allow for a less advanced perturbation methods. Han et al. (2013) draw a line between physical and visual plausibility naming further factors improving the visual plausibility of a scenario such as the number of simultaneous collisions or homogeneity of colliding objects. In physical systems, situations that are non-intuitive to human can occur due to an unobservable state of the environment, e.g an object colliding with a fast rotating wheel or an unexpected behavior of a compressed spring. This opposes the goal of computer graphics were visual plausibility becomes a stronger requirement (Barzel et al., 1996) and motivates the search for more sophisticated methods for sampling probable states in physics engines.

Multiple techniques allow neural networks to incorporate uncertainty in model predictions. Mean-Variance-Estimation (Nix & Weigend, 1994) is a method that circumvents point estimates in the output space by directly predicting a normal distribution. Um et al. (2018) used this method along with a particle-based representation for splash prediction. Assuming independent velocity distributions for each splash particle produced visually pleasing results. In our initial experiments for stochastic simulations, this method lead to unsatisfactory results. We observed that Mean-Variance-Estimation method is capable of indicating highly uncertain situations e.g collisions or force applications. Unfortunately, due to the lack of space-time consistency between particles that are present in real objects, this approach lead to incorrect shape predictions during test time.

Stochastic regularization as a way of capturing model uncertainty is an active field of research with scarce theoretical foundations. Nonetheless, we see a growing number of practical applications of this group of algorithms in numerous research areas (Gal et al., 2017; Bhattacharyya et al., 2017; Kendall et al., 2015; Kampffmeyer et al., 2016). Gal & Ghahramani (2016) proposed applying dropout during training and inference as a Bayesian inference approximation with prediction variance as the measure of the epistemic uncertainty. A clear advantage of this method is the ability to visualize the results of each predicted trajectory. On the other hand the computational cost grows linearly with the number of samples.

Prior work has shown that injecting noise into neural networks is successful not only as a regularization method (Noh et al., 2017; Kang et al., 2016) but also in the training of RL agents Xia et al. (2018); Chua et al. (2018). In model-free RL, temporal credit assignment, sparse reward, and exploration-exploitation trade-offs present significant challenges. Long episodes amplify both problems of credit assignment and reward sparsity, where naive exploration causes exponentially growing sample inefficiency (Osband et al., 2016). Reward shaping is one counter measure that improves credit assignment, and consequently sample efficiency, in model-free RL (Grześ, 2017; Zou et al., 2019). Designing shaping functions usually requires expert knowledge and hand-engineering, while also imposing constraints on how the agent solves the task. Such constraints may prevent the agent from solving the task optimally. Predicting a set of trajectories can be framed here as a reward relaxation method with the clear advantage of depending on a single parameter - dropout rate. Fortunato et al. (2017) introduced parametric noise learned with gradient descent into action prediction network, which led to significantly better exploration and higher rewards without creating large computational overhead. This method shows clear potential for stochastic methods to improve training efficiency in reinforcement learning.

## 3 APPROACH

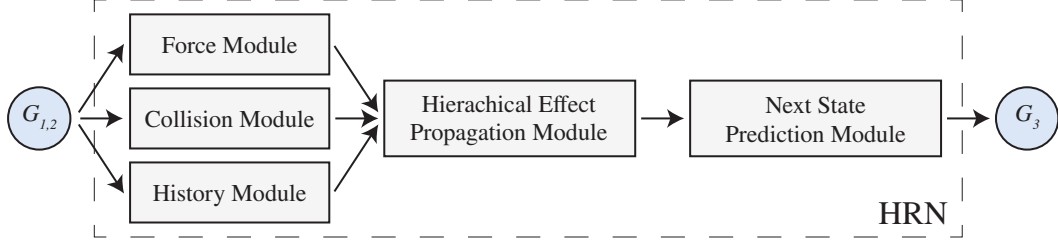

Figure 2: **Hierarchical Relation Network (HRN) architecture.** Force, collision and past effects on particles are computed and then propagated through each object hierarchy. The propagated effects are used to predict the next particle positions. Gray blocks represent graph convolutional effect propagation modules.

Our stochastic forward dynamics model is based on the deterministic hierarchical relation network (HRN) as proposed by Mrowca et al. (2018) and depicted in Figure 2.

**Hierarchical graph representation:** Propagating effects through a fully-connected scene graph is computationally infeasible. HRN circumvents this problem by leveraging a tree-like graph form that defines a constrained subset of edges for more efficient effect propagation. Edges within each object graph comprise shape and material properties, describing how rigid, soft and "cloth-like" the material is as defined in our simulator. Edges across object graphs describe physical relations between objects such as contact forces. The hierarchy construction starts with the particles at the

lowest level as provided by our ground truth simulator. Each next level of the hierarchy is constructed by clustering particles based on their states. The state attributes are the position, the velocity and the mass. Finally, nodes at each hierarchical level represent the state of an object, an object part, an object subpart and so forth down to single object particles. HRN takes a sequence of the past two hierarchical physics graphs $G_{1,2}$ which consist of hierarchical object graphs as input and predicts the next state of the scene graph.

**Force, Collision and History Modules:** HRN assumes three effects that act on particles at the lowest level of the hierarchy and that influence the next particle state: external forces, interactions with particles of other objects in close proximity and particle's state history. Inputs to these modules are current particle states, external forces (Force Module) and states of particles with which the particle interacts (Collision Module). Each module computes an embedding vector that represents the effects of the external forces, the collisions and the past states on the particle using pairwise graph convolutions. These effects are combined by summation and enter the Hierarchical Effect Propagation Module.

**Hierarchical Effect Propagation Module:** HRN predicts the next graph state by estimating the influence that particles within each object and across objects exert upon each other. The compounded effects from the Force, Collision and History Modules are propagated not only up and down the hierarchy, but also between particles at the same level by hierarchical graph convolutions that are implemented as fully-connected networks with weight sharing. Each forward pass takes the states of two connected particles and outputs an embedding vector that describes the influence of the first particle on the latter. These effect embedding vectors are collected for each particle in the scene graph and are used to estimate the future particle velocity.

**Next State Prediction Module** This module is realized as a feed-forward fully-connected neural network and predicts per-particle future velocity. It takes in the current particle state consisting of the position, velocity and mass, as well as the sum of effect embedding vectors belonging to the particle. The predicted particle velocity is expressed in the local coordinate frame, i.e relative the corresponding particle at the higher level in the hierarchy. The state of the particle at the highest level additionally includes gravity and is defined in the global coordinate frame.

To make the HRN stochastic and sample multiple plausible trajectories for a given initial state, we introduce a Monte Carlo based dropout on the activations of the graph convolutional collision and force modules. We greatly improve model predictions by introducing a new fully-connected shape loss term and a state-invariant recurrent training procedure. Our final model creates realistic trajectories that are suited to train a model-free agent for manipulation tasks.

## 3.1 STABLE LONG PREDICTIONS VIA RECURRENT TRAINING AND SHAPE CONSTRAINTS

Stable realistic long-term predictions are crucial for planning tasks. While HRN predictions are accurate for a large number of complex physical scenarios, we identify that objects fall apart relatively quickly along boundaries of object parts for two reasons.

First, the HRN's loss function is designed to minimize the error between predicted and ground truth states while imposing a group shape constraint. As shown in the left panel of Figure 3, this group shape loss optimizes the pairwise distances between object nodes within a group to be the same as the ground truth pairwise distances. It does not impose that the pairwise distances across groups are the same as the ground truth distances, which leads to unrealistic deformations between groups as depicted in Figure 5. We therefore introduce a stronger fully-connected shape constraint (Figure Figure 3, right) that imposes pairwise distances to be the same as ground truth pairwise distances across all possible node combinations, which improves shape preservation significantly.

Second, the HRN's prediction errors accumulate exponentially as predictions are fed back in recurrently during inference to generate multi-step trajectory predictions. However, during training the HRN is only supervised with the next ground truth state and thus never gets its own perturbed predictions as input. To make the HRN robust against prediction errors, we therefore propose to train the model recurrently in a state-invariant way, i.e. without using a hidden state as physical dynamics is state-free (Figure 4). The overall loss is the sum of losses from each time step. Learning recurrently on long sequences, the network optimizes its weights taking into account its own prediction errors during training. This significantly reduces error accumulation during inference time.

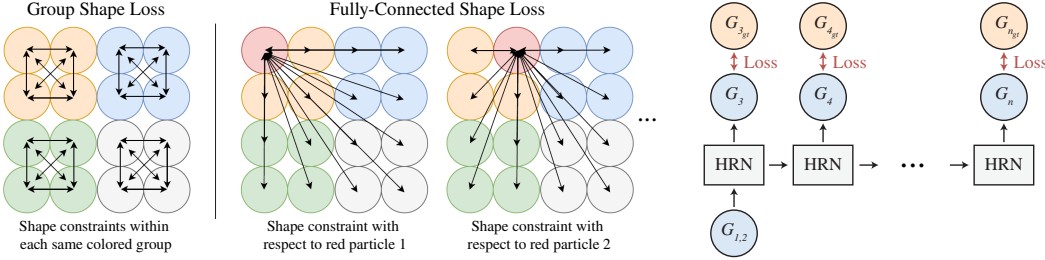

Figure 3: **Shape loss.** The HRN shape loss only constraints particle distances within object particle groups (left). Our new fully-connected shape loss constraints distances between all particles pairs within each objects (right).

Figure 4: **Recurrent training.** Model predictions are fed back recurrently as input to stabilize long-term predictions.

## 3.2 SAMPLING REALISTIC TRAJECTORIES WITH MONTE CARLO DROPOUT

For sampling physically plausible object trajectories, we introduce a novel Monte Carlo sampling based graph-convolutional dropout method. Dropout as proposed by Srivastava et al. (2014) removes a certain number of randomly chosen nodes in a neural network to prevent overfitting with a probability commonly referred to as dropout rate. In each iteration, a new set of nodes is sampled and only the edge weights attached to the active nodes are updated via backpropagation. In our novel approach, we use randomly sample dropout masks on graph-convolution kernels to sample physically plausible trajectories. To keep the kernel fixed independent of its position in the graph, we only sample once per prediction step. To infer a set of plausible trajectories, we randomly sample a dropout mask for each generated trajectory during test time, similarly how Gal & Ghahramani (2016) uses dropout to generate multiple predictions. The modular architecture of the HRN allows us to apply dropout at different locations in a very interpretable way (Figure 2. Dropout on the collision module makes sampled trajectories diverge at collision points. Dropout on the force module leads to diverging trajectories during force applications. Dropout on other HRN modules lead to convergence problems and unrealistic predictions. In the following experiments, we thus only apply dropout to the HRN's force and collision modules. Applying our dropout based sampling method on our dynamics model results in physically plausible long-term predictions with consistent shapes.

## 3.3 MODEL-FREE REINFORCEMENT LEARNING ON STOCHASTIC ENVIRONMENTS

The ability to sample a distribution of physically plausible trajectories can be used to improve the efficiency of exploration of model-free reinforcement learning agents. We thus train a model-free policy on our stochastic physics predictor to achieve physical manipulation tasks as follows. At each episode during training, we input the agent's action and current state into our stochastic forward dynamics predictor and sample a set of 5 future states with our dropout method. For accelerated training, we introduce a reward relaxation method which consists of rewarding an agent as soon as one of the trajectories from the sampled set leads to the goal, naturally exposing the agent to rewards much quicker. If none of the trajectories hits the goal, one future state is chosen at random and the sampling process is repeated for the next future state. The level of reward relaxation is controlled by the dropout rate. The higher the dropout rate, the wider the set of trajectories and the easier it is for the agent to be rewarded. This directs the agent much quicker towards the reward in early training stages. In scenarios requiring high levels of accuracy and repeatability, we find that a gradual reduction of the dropout rate during policy training helps convergence and leads to more efficient policies compared to a fixed dropout rate, which we show in the following experiments.

## 4 EXPERIMENTS

In our experiments, we first show that our recurrent training and new fully-connected shape loss significantly improve the prediction quality for single long-term trajectories on complex physical scenarios over baselines. We then demonstrate how our proposed Monte Carlo sampling based dropout method generates multiple high-quality trajectories by visualizing stochastic model roll-

outs. Lastly, we use our stochastic forward dynamics model's ability to generate multiple trajectories to train a model-free policy on two physical manipulation tasks more efficiently and to higher reward.

### 4.1 FORWARD DYNAMICS PREDICTION PERFORMANCE

#### 4.1.1 EXPERIMENTAL SETUP

We evaluate our models forward dynamics prediction performance against the HRN baseline (Mrowca et al., 2018) on two complex scenarios. The first scenario showcases the ability of our model to predict complex deformations: A deformable soft cube is first lifted off the ground by an upward impulse and then falls toward the ground while rotating and deforming on impact (Figure 5 left). In the second scenario, we evaluate our models performance on complex collisions. Collisions can greatly magnify object position and pose errors leading to large discrepancies between predictions and ground truth. For our collision experiment, two rigid cubes are placed at a random distance from each other and then repeatedly accelerated by an impulse on each cube into another to generate collisions (Figure 5 right). We train our model on a multitude of examples of both scenarios and evaluate on held out examples. We compare the mean squared error on positions, velocities and shape loss and show qualitative long-term predictions of our model and the HRN baseline.

#### 4.1.2 RESULTS

In Table 1, we present our quantitative results on the deformation and collision task. Our fully-connected shape loss and recurrent training procedure significantly lower long-term prediction errors in both scenarios. On the collision task, initial position and velocity error increase slightly compared to the baseline but accumulate to far lower errors in the long run. Empirically, we found that our recurrent training procedure works best with sequence lengths between 4 and 6 time steps. Longer sequence lengths prevent the model from converging during training. We found that gradually increasing the sequence length during training is an effective countermeasure.

The improvements of our model over the HRN baseline are obvious in visualizations of predicted trajectories (Figure 5). Whereas HRN predictions fall apart along object boundaries and sometimes penetrate objects, our method preserves shapes and resolves collisions much better and predicts positions much closer to the ground truth, leaving us with an adequate basis for generating multiple plausible trajectories with our sampling method.

Table 1: **Quantitative dynamics prediction evaluation**. We compare the mean squared error on positions, velocities and shape preservation for our model with the baseline HRN model. Our model outperforms the baseline on deformation predictions and long-term collision predictions.

|  |  | Position MSE | | Velocity MSE | | Shape MSE | |
|---|---|---|---|---|---|---|---|
|  |  | t+5 | t+15 | t+5 | t+15 | t+5 | t+15 |
| **Deformations** | HRN | 0.00064 | 0.017 | 0.00019 | 0.0013 | 0.037 | 0.092 |
|  | Ours | **0.00039** | **0.011** | **0.00003** | **0.00043** | **0.0051** | **0.019** |
| **Collisions** | HRN | **0.028** | 0.84 | **0.0045** | 0.032 | 0.018 | 0.054 |
|  | Ours | 0.029 | **0.63** | 0.0061 | **0.019** | **0.0020** | **0.0039** |

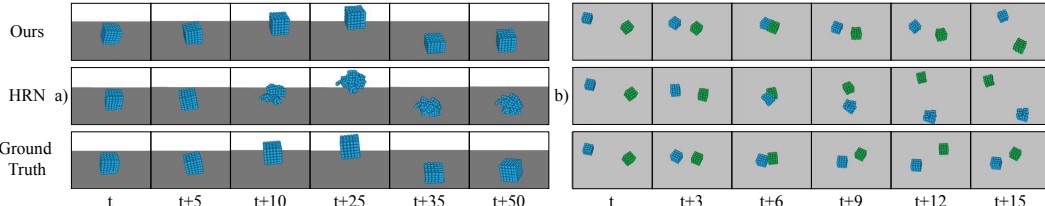

Figure 5: **Dynamics prediction comparisons.** Our method is compared to the HRN baseline and ground truth. **a)** A soft cube bounces of the ground. **b)** Two rigid cubes collide. Our method preserves the geometry of objects better over long time horizons.

### 4.2 SAMPLING MULTIPLE PLAUSIBLE TRAJECTORIES

#### 4.2.1 EXPERIMENTAL SETUP

In this section, we demonstrate that our Monte Carlo sampling based dropout method can sample multiple physically plausible trajectories from our forward dynamics predictor under the same initial state. In two complex scenarios we study our model's uncertainty during force applications and collisions. In the first scenario, an external force lifts a soft body, which subsequently drops toward the floor rotating slightly. By applying dropout to the force module (Figure 2), we generate multiple trajectories that arise due to our model's uncertainty during force applications. We use a dropout rate of 0.1 during training and 0.3 during testing. In the second scenario, we show that our proposed method produces realistic sets of trajectories in collision scenarios. Forces are applied to two rigid cubes pushing them towards each other causing collision. Dropout is applied to both force and collision module with a rate of 0.05 both during training and testing.

#### 4.2.2 RESULTS

The visualizations of multi trajectory roll-outs in Figure 6 show that the sets of predicted trajectories are physically and visually plausible. Due to the modularity of the HRN model, targeted stochasticity can be applied within each submodule via dropout, introducing uncertainty in the output of force and collision predictions. Our proposed sampling method is able to capture trajectory distributions ranging from single mode low variance to complex, multi-modal distributions. Dropout rates between 0.05 and 0.3 allow for fast convergence during training and a wide variety of visually plausible sample trajectories during inference. We notice that inference dropout rates that differ significantly from the training rates can cause biased predictions leading to, e.g. objects slowly drifting away in one direction. Additional results studying the effect of the dropout rate on the width of the state distributions can be found in supplement Figure 10.

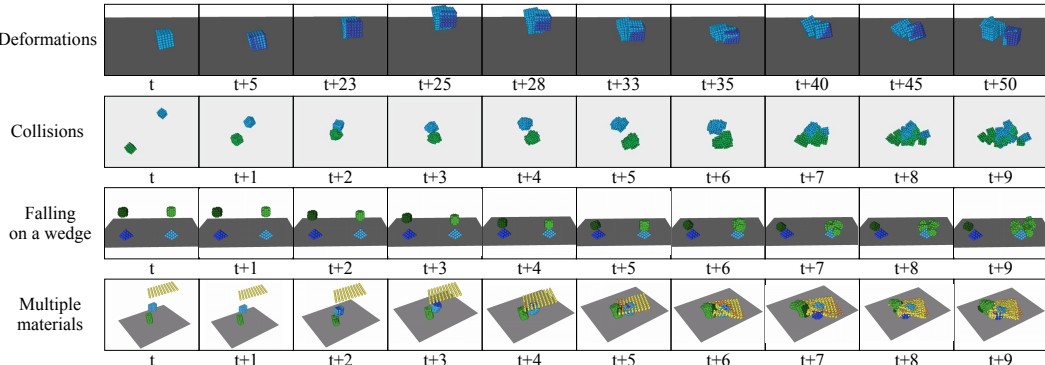

Figure 6: **Example sampled multiple trajectories.** We use dropout on the force and collision module to sample multiple trajectories given the same initial input. Dark colors depict the ground truth trajectory. Light colors depict imagined sampled trajectories. The performance of our model in forward simulation is maintained at a high level despite the introduced graph-convolutional dropout. **Deformations:** A soft cube bounces off the ground. **Collisions:** Two rigid cubes collide. Our method is able to sample multiple physically plausible trajectory in each scenario. **Falling on a wedge:** We simulate the situation from Fig 1. A rigid objects falls on a wedge. Stochastic simulation allows for a multi-modal prediction. **Multiple materials:** A rigid and a soft object interact with a cloth while falling on the ground.

### 4.3 MODEL-FREE REINFORCEMENT LEARNING

#### 4.3.1 EXPERIMENTAL SETUP

We show that stochastic physical environments are useful for intelligent systems by training reinforcement learning agents in two different scenarios involving various physical interaction types and materials. We use Proximal Policy Optimization (Schulman et al., 2017) as a model-free reinforce-

ment learning method in all scenarios. Similarly to Plappert et al. (2017), we add a further baseline in which we use the deterministic environment and add Gaussian noise in the action space.

**Cube moving task:** In this scenario, the agent learns to apply a sequence of forces to rigid cubes such that at least one of the cubes is pushed towards the goal region. The maximum length of the episode is 10. Here, the two cubes are transparent to each other and cannot collide. The stochasticity originates entirely from the uncertainty in the force application. We use a constant dropout rate of 0.1 in the force module of our physics predictor throughout the whole training.

**Ball hitting tower task:** Tasks that require inducing collisions pose a significantly more difficult challenge to the RL agent. In this scenario, there is a stack of three rigid cubes and a ball to which the agent can apply force. The agent gets a reward for pushing the middle cube out of the tower. To achieve the goal, the agent needs to hit the tower with the ball to which it applies forces. The maximum episode length is 15 steps.

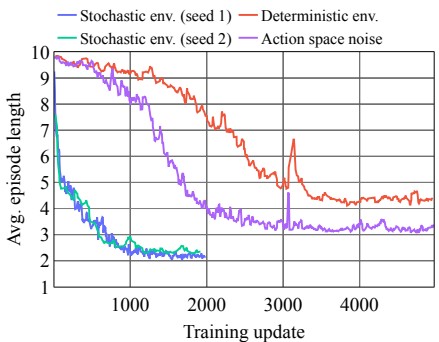

Figure 7: **Average episode length in the "cube moving task".** We compare a deterministic environment against action space noise and 2 randomly seeded stochastic environments. The agent learns faster in stochastic environments through better initial exploration and converges to a shorter policy.

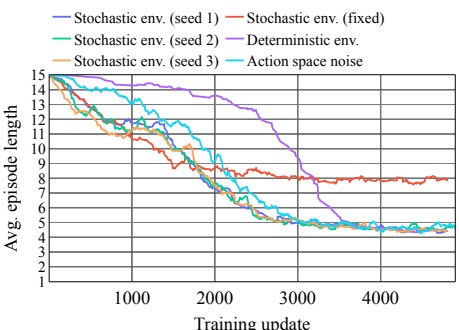

Figure 8: **Average episode length in the "ball-hits-tower task".** We compare a deterministic environment against action space noise, a stochastic environment where the dropout rate is fixed and 3 randomly seeded stochastic environments where the dropout rate is annealed. The agent finds shorter policies earlier in the training indicating more efficient exploration in the stochastic environments.

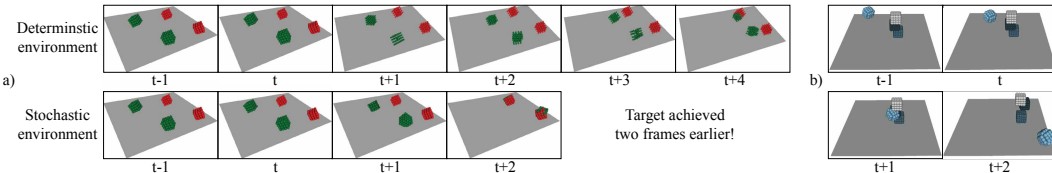

Figure 9: **Policy comparisons. a) Cube moving task. Top**: Policy learned in a deterministic environment (longer, 4 time steps). **Bottom**: Policy learned in a stochastic environment (shorter, 2 time steps). The first two frames are model inputs. The red cubes indicate the target position to which the green cubes have to be moved. **b) Ball hitting tower task.** Agents in deterministic and stochastic environments converge to similar 4 step policies. The figure depicts one 4 step policy example.

### 4.3.2 RESULTS

**Cube moving task:** In this example, introducing the action space noise leads to faster learning compared to learning in the deterministic environment. Training in stochastic physical environments outperforms both baselines, allows for better exploration and finding of shorter policies. In Figure 9, we visualize the two learned policies, in stochastic and deterministic environments.

**Ball hitting tower task:** In this scenario, the agent finds more efficient policies through the application of stronger forces during training, which results in gradually shorter policies as shown in Figure 8. The most effective learning method is learning in a stochastic environment with the dropout rate annealing. We lower the dropout rate linearly from 0.1 at the start to 0 after 1200 training updates. This method allows for initial fast exploration, but does not introduce too high stochasticity when precision is needed as the agent begins applying strong forces later in the training. Without annealing, the randomness is too high and agents learn longer policies in a noisy training process, as indicated by the red curve in Figure 8. Furthermore, action space noise improves the pace at which the agent learns compared to the entirely deterministic environment. The policies learned by the presented methods do not significantly differ. An exemplary policy is visualized in Figure 9.

## 5 Conclusion

Qualitatively our stochastic HRN predicts plausible future trajectories; an experiment in which human subjects were asked to discriminate between ground-truth and predicted trajectories could be used to evaluate its performance quantitatively. Even though this method does not require extensive expert knowledge, a few design decisions have to be made e.g dropout rates for training and inference. During inference, too high of a dropout rate can lead to visually unrealistic dynamics and object interactions. Dropout rate scheduling during training should be investigated to improve convergence of the dynamics model during training, which may improve its performance as an environment for the reinforcement learning tasks. Possible optimizations include more complex, potentially non-linear, annealing schedules during inference, delaying the dropout rate annealing, and finding appropriate starting values. Finding a universal schedule that can be applied to any environment and task has large potential for accelerating reinforcement learning. Further improvements for the physics predictor are key for its use as a physical environment. These can include improvements for: scenarios with multiple materials in one scene, penetrations during collisions that can lead to insufficient position prediction, and generalization to new scenes.

Our results show that the proposed sampling method produces physically plausible trajectories in single- and multi-object scenarios as well as across a range of materials. The quality of roll-outs, e.g. shape prediction is not compromised by the introduced noise. Furthermore, our model-free reinforcement learning experiments indicate that agents learning in physically stochastic environments are able to explore better and learn quicker, which confirms the quality of the sampled trajectories. In difficult reinforcement learning scenarios, where a high level of precision is needed to get a reward, we demonstrated that dropout rate annealing is an effective method to avoid too high randomness at the same time not reducing the benefits of stochasticity for exploration in early stages of the training. In this regard, stochastic neural physics engines offer a clear advantage over conventional physics engines.

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

# 6 APPENDIX

## 6.1 EFFECT OF DROPOUT RATE ON THE WIDTH OF THE TRAJECTORY DISTRIBUTIONS

In Figure 10, we present the effect of the dropout rate on the width of the predicted distribution of trajectories. In this example, dropout is activated in the force module. We compare distributions generated with dropout rates 0.5, 0.3 and 0.1. In Figure 11, we present the stochastic predictions in "ball hitting tower" scenario. The dropout is active in the force and collision modules. The prediction variance corresponds to the dropout rate strength in both scenarios. In the "ball hitting tower" scenario, we observe highly complex behavior, where the time at which the cubes fall off the tower at different rates

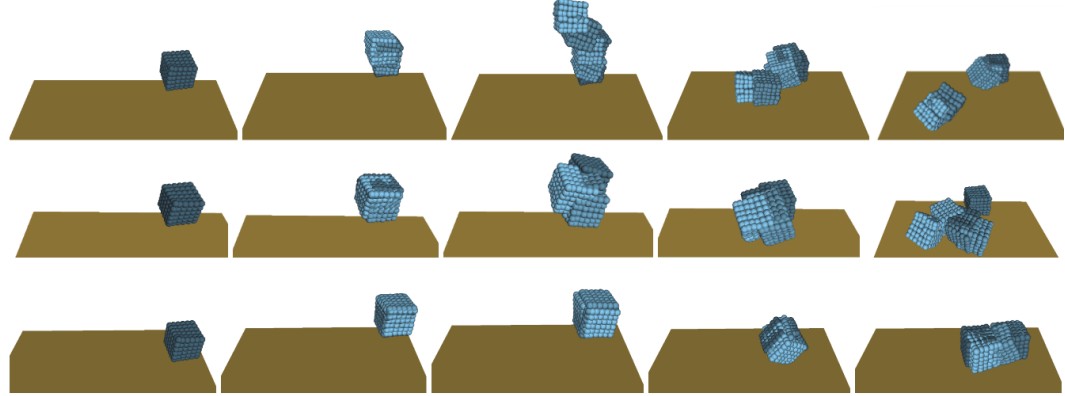

Figure 10: Effect of the dropout rate on the distribution width of the sampled trajectories. 0.5 - top, 0.3 - middle, 0.1 - bottom. Dropout applied in the force module.

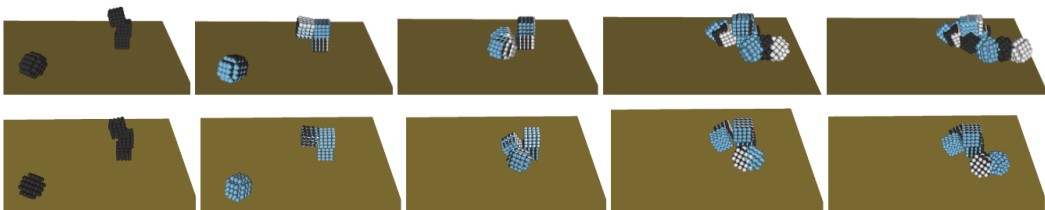

Figure 11: Effect of the dropout rate on the distribution width of the sampled trajectories. 0.3 - top, 0.1 - bottom. Dropout applied in the force and collision modules.

