# OpenReview forum: "Stochastic Neural Physics Predictor"
_ICLR.cc/2020/Conference — Reject_

### Official Review · AnonReviewer2 · 2019-10-22
**Official Blind Review #2**

**Rating:** 3

**Review:**

Overview:
This paper introduces a method for physical dynamics prediction, which is a version of hierarchical relation network (Mrowca ‘18). HRNs work on top of hierarchical particle-based representations of objects and the corresponding physics (e.g. forces between different parts), and are essentially are graph-convolutional neural networks.
Unlike the original work, the proposed method introduces several improvements: 1. an updated loss function, that adds distance constraints between *all* the particles of the object. 2. recurrent training, when the predictions are fed to the inputs. 3. adding dropout.

Writing:
The paper is relatively well-written and easy to follow.

Evaluation:
Authors compare their model on a dynamics prediction task and seem to outperform the original HRN, especially on longer-term sequences. In addition they report results for trajectory sampling (qualitative) and model-free RL, where using their model as a stochastic simulator seems to have positive impact on agent training.

Decision:
Although the proposed improvements upon HRN generally make sense, it is not clear if those are very significant on their own: adding dropout and recurrent training do not seem particularly novel and, since there is no ablation study, it is hard to see what exactly contributes to the reported improvements.
As for the experimental evaluation, it seems like important baselines are missing, and the model seems to be very sensitive to hyperparameters (see questions). Thus, I am currently leaning more towards a rejection, hence the “weak reject” rating.

Various questions / concerns:

* It is not clear why authors do not provide a comparison to DPI-Nets (Li ICLR‘19). This model seems to be outperforming HRN, and from what it looks like is publicly available: https://github.com/YunzhuLi/DPI-Net. I would encourage authors to provide comparison to this baseline, and potentially on similar sets of experiments, or explain why this comparison would not be possible (which seems unlikely).

* Authors seem to acknowledge that the model is sensitive to the hyperparameter choice (dropout rates), however, there is no numerical evaluation that would help readers understand how critical this choice is for the final performance. Judging from very specific settings in different experiments, this could be a serious concern.

* I find it a bit strange that results in Fig. 7-8 are for random seeds. Is it not possible to just plot an average for e.g. 10 runs?

Update:
I would like to thank the authors for a detailed response!
It seems like there is a common concern about the novelty among reviewers: improvements over HRN are quite incremental. Although authors provide a verbal justification for not comparing to another strong baselines, I do not see why would it not be possible to compare methods in the similar settings, even though that baseline might be more limited.
Generally, if the main contribution is actually only the "stochastic" part and not improved performance, then just adding dropout does not seem like a particularly novel approach to me: whether the convolutions are on graphs or on euclidean domains, this does not change the way dropout is done.












**Experience Assessment:**

I have read many papers in this area.

**Review Assessment: Checking Correctness Of Derivations And Theory:**

I carefully checked the derivations and theory.

**Review Assessment: Checking Correctness Of Experiments:**

I assessed the sensibility of the experiments.

**Review Assessment: Thoroughness In Paper Reading:**

I read the paper thoroughly.

---

> ### Author Response · Authors · 2019-11-14
> **Response to Reviewer #2**
>
> Thank you so much for the comments!
>
> “It is not clear why authors do not provide a comparison to DPI-Nets (Li ICLR‘19). This model seems to be outperforming HRN...”
>
> We have considered this architecture, and there are several reasons why we found the comparison to DPI-Nets not crucial for our work. To begin with, it is questionable if the DPI-Nets model clearly outperforms the HRN. HRN aims at solving a more difficult and general problem. As HRN works across different materials without special material constraints, we choose to work with the HRN instead of DPI net in our work. In rigid body simulations, DPI-Nets assumes rigidity of objects and predicts only rotation and translation, which is a different and much easier task. For rigid bodies this requires access to the undeformed ground truth shape over the whole prediction sequence. This contrasts with HRN, which fully learns the dynamics of each material without specialized constraints and does not require the ground truth shape to be provided at every prediction step. Our proposed model can predict correct dynamics and shapes in complex scenes without these constraints on par with DPI-Nets. A specific example is our scene involving three different, interacting objects, where all three are made from different materials: rigid, deformable and cloth. We also would like to point out that our experiments are at least on par with the complexity of the rigid fluid experiments used by Li et al.. Unnatural deformations for rigid and elastic soft bodies and cloth are much more salient to the human observer than wrong predictions of single fluid particles. We encourage reviewers to consider the following video, where we present our results of the simulations from the paper as well as additional more complex scenarios, which will be added to our manuscript: https://www.dropbox.com/s/59dalr767rlqhwm/Stochastic_Neural_Physics_Predictor_Experiments.mp4?dl=0
>
> Furthermore, the main focus of our work is finding ways of introducing stochasticity for simulating physical dynamics that reliably produce visually plausible results. The work of Li et al. did not explore this area, which makes the two works orthogonal in this regard. Nonetheless, DPI-Nets can potentially also benefit from our findings, both in terms of modelling stochasticity and improving prediction quality.
>
> “Authors seem to acknowledge that the model is sensitive to the hyperparameter choice (dropout rates)...”
>
> We included qualitative example rollouts for several different dropout rates in the Appendix. It is hard to find a good quantitative metric that captures the quality of rollouts as well as actual examples. Smaller perturbations might capture the mean better, but capture fewer possible modalities than larger perturbations, and the types and magnitude of perturbations might not transfer across different situations. Compared to input force and position perturbations, our dropout method significantly simplifies this problem to one parameter. For example with conventional perturbation methods, if we would like to perturb the simulation for colliding rigid cubes, it is not possibly to simply perturb each single particle. Instead, we need to rotate and translate the particles in accordance with the entire shape of the object such that it doesn’t deform unnaturally. For a soft object it would be even harder to hard-code plausibly looking perturbations. With our dropout method we only have to adjust one parameter to control how widely sampled trajectories are spread which is a huge simplification over previous methods. We find an even more constrained set of values between 0.7 and 0.9 to work across a wide range of scenarios. The attached video further illustrates the problem of hard coded input perturbations and the advantage of learned dropout perturbations which flexibly and automatically adapt to different scenarios:  https://www.dropbox.com/s/59dalr767rlqhwm/Stochastic_Neural_Physics_Predictor_Experiments.mp4?dl=0 (at 0:48)
>
> “I find it a bit strange that results in Fig. 7-8 are for random seeds. Is it not possible to just plot an average for e.g. 10 runs? “
>
> There is a clear intent behind plotting the curves separately. The training process in model-free RL is in most cases very sensitive not only to hyperparameters, but also the random seed. Our goal was to clearly show that agents learning in stochastic physical environments learn faster and converge to better policies than our baselines independent of the chosen random seed. Henderson et al. [1] showed that the variance between runs with different random seeds is sufficient to drastically change the resulting distributions hiding the true performance of RL algorithms. Hence, plotting the mean over multiple runs can be misleading.
>
> [1] https://arxiv.org/pdf/1709.06560.pdf

---

### Official Review · AnonReviewer1 · 2019-10-23
**Official Blind Review #1**

**Rating:** 3

**Review:**

Summary

There has been work on deep learning based forward dynamics model to learn the dynamics of physical systems. In particular, the hierarchical relation network (HRN) as proposed by Mrowca et al. (2018). However, HRN is deterministic. This is problematic for long-term predictions because of the uncertainty of physical world. This paper builds on top of HRN. It proposes a Monte Carlo sampling based graph-convolutional dropout method that can sample multiple plausible trajectories for an initial state given a neural-network based forward dynamics predictor. It also introduces a shape preservation loss and trains the dynamics model recurrently to better stabilize long-term predictions. It demonstrates the two techniques improve the efficiency and performance of model-free reinforcement learning agents on several physical manipulation tasks.

Strengths

Learning physics models which accounts for the multi-modal nature of the problem is very important. Graph-convolutional dropout is one method to deal with the multi-modal nature of the problem. It is a very good contribution.

 The tasks evaluated are not very sophisticated.

Weaknesses

The paper's contribution is very incremental.

There should be a discussion on the different type of methods to account for uncertainties, e.g. bayesian neural networks and how they differ in terms of multi-modal predictions.

Because the tasks are not very complicated, it is not clear how good the whole neural physics predictor is.


**Experience Assessment:**

I have read many papers in this area.

**Review Assessment: Checking Correctness Of Derivations And Theory:**

I carefully checked the derivations and theory.

**Review Assessment: Checking Correctness Of Experiments:**

I carefully checked the experiments.

**Review Assessment: Thoroughness In Paper Reading:**

I read the paper thoroughly.

---

> ### Author Response · Authors · 2019-11-14
> **Response to Reviewer #1**
>
> Thank you so much for the comments!
>
> “The paper's contribution is very incremental. “
>
> The use of dropout in graph convolutional networks for simulation of multi-modal problems is a simple yet elegant solution that is new to the best of our knowledge. We are glad that it has been mentioned as a “very good contribution” by the reviewer. We agree that dropout itself is not a novel method. However, a naive implementation in graph convolutional networks has a negative impact on predictions. In our work, we proposed a novel and effective implementation in graph convolutional networks that does not lower system performance. In our experiments, it does not cause object shape degradation that is present when dropout is applied in a conventional way, as we discuss in Section 3.2.
>
> The training process in reinforcement learning (RL) is sensitive to hyper parameters, often also to a random seed. Overparameterized systems are difficult to work with, due to e.g. computationally expensive hyperparameter searches, which is a reason why our proposed easy-to-use method for improving the training in model-free RL has the potential to be adopted on a wider scale. Our method can also be seen as a reward relaxation method. Due to its high effectiveness for a very common problem in RL, this contribution should not be underrated. Our proposed training method significantly outperforms other perturbation methods, e.g., commonly used action space noise, offering a promising future direction for RL community.
>
> “There should be a discussion on the different type of methods to account for uncertainties...”
>
> As mentioned in Section 2, we explored Mean-Variance-Estimation method [1] for predicting plausible object trajectories. At test time, sampling from independent per-particle normal distributions failed due to a lack of space-time consistency between object particles that are present in real objects. We will make this point clearer and underline that such an experiment has been conducted. We also evaluated this method as an uncertainty estimator, which gave reasonable results e.g indicating relatively high prediction uncertainty during collisions and force applications. We are happy to provide the results, although we decided to not include them in our original submission. As proposed by [2] Monte Carlo Dropout can be seen as a Bayesian approximation for inference. With this definition, it falls under the category of Bayesian Neural Networks. Furthermore, Bayesian neural networks are part of a group of algorithms that exhibit limited scalability, and as such are not applicable to large systems like ours. This is one of the motivations for our implementation via Monte Carlo dropout as a viable solution.
>
> “Because the tasks are not very complicated, it is not clear how good the whole neural physics predictor is.”
>
> Regarding concerns about experiments not being complicated enough, we would like to argue that
> our selected scenarios include several highly complex geometries, physical effects (collisions, object deformations) and materials (rigid, deformable/soft, cloth). Most recent papers in the intuitive physics domain still deal with simple 2D scenarios [3]. However, to show that our model works on a greater set of scenarios, we will include more examples in the revised version and we are happy to include more scenarios on acceptance which we could not include because of rebuttal time constraints.
>
> As a response, we conducted more highly complex experiments in an extended set of scenarios, including new geometries and materials. We strongly encourage to watch the following video with visualizations of our results:
> Scenario 1: In “Multiple Materials”, three distinct geometries, each made of different materials (soft body, rigid body, cloth) interact in one scene. They are lifted, drop on the floor and collide with each other. In a stochastic simulation, the shapes are preserved well, positions are well distributed with respect to the ground truth and the trajectories are visually plausible.
> Scenario 2:  In “Ball hitting tower” (at 1:05 of the video), we compare our stochastic simulation method to simple input perturbation, but the performance of the physics predictor in a forward simulation can be assessed. This task includes a stack of 3 cubes, which are hit by a ball. Stack is a challenging case of a very frequent collision, which our model predicts well.
>
> Scenarios 1 and 2: https://www.dropbox.com/s/59dalr767rlqhwm/Stochastic_Neural_Physics_Predictor_Experiments.mp4?dl=0
>
> Scenario 3: We simulated the toy experiment presented in Figure 1. A ball falls on top of a pyramid.
> This experiment shows a clear advantage of a stochastic physics engine over deterministic ones.
> https://www.dropbox.com/s/btpkvgtf84zea8l/Stochastic%20Neural%20Physics%20Predictor%20Experiment%202.mp4?dl=0
>
> [1] https://ieeexplore.ieee.org/document/374138
> [2] https://arxiv.org/pdf/1506.02142.pdf
> [3] https://arxiv.org/pdf/1904.03177.pdf

---

### Official Review · AnonReviewer3 · 2019-10-23
**Official Blind Review #3**

**Rating:** 6

**Review:**

The authors describe a neural architecture with dropout randomizer with the aim of producing an ensemble of physical trajectories that seem plausible for a human eye. The question is relevant and interesting.

The authors bring in two improvements to the HRN model by (Mrowca et al., 2018). They drop the hierarchy from the shape loss, which is decipeted in Figure 3, and provide a change in the recurrent training decipited in Figure 4.  Figure 4 is only showing the modified training, not the initial HRN training, although Figure 3 contains both. For the sake of consitency and good read the Figure 4 should contain the original training image.

The description of the system is very verbose, but a concrete description of the system is only in the references, which make the manuscript hard to read and it does not fit well in a conference where one cannot assume that the audience is not an expert of this particular subfield.

As a remedy, I would suggest, a simple one-hieararcy level architecture with hyperparameters to show what is really happening. One could use the system in Figure 1 as a more thorough example.

The results looks good and valuable, altough the images provided are quite small.















**Experience Assessment:**

I have published in this field for several years.

**Review Assessment: Checking Correctness Of Derivations And Theory:**

I assessed the sensibility of the derivations and theory.

**Review Assessment: Checking Correctness Of Experiments:**

I assessed the sensibility of the experiments.

**Review Assessment: Thoroughness In Paper Reading:**

I read the paper thoroughly.

---

> ### Author Response · Authors · 2019-11-14
> **Response to Reviewer #3**
>
> Thank you so much for the comments!
>
> “The description of the system is very verbose, but a concrete description of the system is only in the references, which make the manuscript hard to read and it does not fit well in a conference where one cannot assume that the audience is not an expert of this particular subfield.“
>
> We apologize and understand that the description was not detailed enough. We will correct it in the new version of the manuscript.
>
> “As a remedy, I would suggest, a simple one-hieararcy level architecture with hyperparameters to show what is really happening. One could use the system in Figure 1 as a more thorough example.”
>
> In response, we simulated the situation presented in Figure 1 and included results in the attached video. A rigid ball falls on on top of a pyramid, bounces off and falls on the ground. Stochastic simulation clearly shows the ability of the proposed system to predict multiple plausible trajectories, while maintaining object shape. This shows that the introduced perturbation in the system in form of graph convolutional dropout is of appropriate type and magnitude for the situation. (Video with results: https://www.dropbox.com/s/btpkvgtf84zea8l/Stochastic%20Neural%20Physics%20Predictor%20Experiment%202.mp4?dl=0)
> The ablation study for different hierarchy parameters has been conducted in [1] in Appendix Section C. As we kept the original implementation of the hierarchy; we claim that these findings are applicable to our model.
>
> “The results looks good and valuable, although the images provided are quite small.”
>
> We will correct the figures in the new version of the manuscript. We encourage you to consider the following high resolution video that includes the experimental results from the paper as well as additional material: https://www.dropbox.com/s/59dalr767rlqhwm/Stochastic_Neural_Physics_Predictor_Experiments.mp4?dl=0
>
> If you have further comments or questions regarding the paper, please do not hesitate to contact us.
>
>
> [1] Damian Mrowca, Chengxu Zhuang, Elias Wang, Nick Haber, Li Fei-Fei, Joshua B Tenenbaum, andDaniel LK Yamins. Flexible neural representation for physics prediction. InAdvances in NeuralInformation Processing Systems, 2018.

---

### Author Response · Authors · 2019-11-14
**General response**

We want to thank all reviewers for their feedback. Our work was well received, addressing the very relevant and interesting problem of plausible multi-modal outcomes of intuitive physics scenarios (R1-3) with graph convolutions dropout as a “very good contribution” (R1) and recurrent training and shape preservation as contributions to stabilize long-term predictions (R1-3). The results look good and valuable (R3), outperforming baselines especially on long-term predictions (R2). The final fully trained stochastic simulator has a positive impact on the training of model-free RL agents for physical manipulation tasks (R1-2). The paper is well written and easy to follow (R1) with weaknesses in the description of the system (R3) which will be addressed in the revised version by incorporating improvements in figures and text suggested by the reviewers. We have also added further experiments and attached high resolution videos in individual responses to address specific concerns of the reviewers about e.g. figure quality (R3), complexity of the experiments and overall performance of our physics predictor (R1, R3).

All in all, we hope that this illustrates that our contributions are far beyond incremental (R1, R2). Rather, they are a significant step towards fully learned, flexible intuitive physics models that are able to forward simulate complex multi-model physical systems with multiple possible outcomes. Using dropout on graph convolutions to generate multiple plausible trajectories is a simple yet elegant solution and has to our knowledge not been done before. The combination of improved shape loss and recurrent training greatly stabilizes long term predictions to be on par or better than methods which require specialized constraints and access to ground truth shapes. This in turn leads to greater flexibility across materials and different scenarios. We apologize if we didn’t do a good enough job in pointing out the differences to prior work before the rebuttal, but we have clarified these points and will include reviewer suggestions in our revised and improved manuscript.

---

### Decision · Program_Chairs · 2019-12-19

**Decision:**

Reject

**Comment:**

The paper presents a timely method for intuitive physics simulations that expand on the HTRN model, and tested in several physicals systems with rigid and deformable objects as well as other results later in the review.

Reviewer 3 was positive about the paper, and suggested improving the exposition to make it more self-contained. Reviewer 1 raised questions about the complexity of tasks and a concerns of limited advancement provided by the paper. Reviewer 2, had a similar concerns about limited clarity as to how the changes contribute to the results, and missing baselines. The authors provided detailed responses in all cases, providing some additional results with various other videos. After discussion and reviewing the additional results, the role of the stochastic elements of the model and its contributions to performance remained and the reviewers chose not to adjust their ratings.

The paper is interesting, timely and addresses important questions, but questions remain. We hope the review has provided useful information for their ongoing research.